# LOOP: A Plug-and-Play Neuro-Symbolic Framework for Enhancing Planning in Autonomous Systems

## Abstract

Planning is one of the most critical tasks in autonomous systems, where even a minor error can lead to significant failures or losses. Current state-of-the-art neural planners struggle in complex domains, often producing plans with missing preconditions, inconsistent goals, or hallucinated steps, while classical planners provide guarantees but lack the flexibility and natural-language understanding needed in modern systems. Existing neuro-symbolic methods typically perform a one-shot translation from natural language to formal plans. In safety-critical autonomous systems, this leaves no mechanism to detect and correct specification errors before execution. To address this, we introduce LOOP, a neuro-symbolic planning framework that models planning as an iterative interaction between neural and symbolic modules. It synthesizes Planning Domain Definition Language (PDDL) models from task descriptions, refines them using feedback from a symbolic planner and execution rollouts, and builds a causal knowledge base from traces to guide subsequent plans. Across six International Planning Competition (IPC) domains, LOOP attains 85.8% task success, surpassing LLM+P (55.0%), LLM-as-Planner (19.2%), and Tree-of-Thoughts (3.3%). Together, these results indicate that consistent planning arises from sustained interaction between neural and symbolic reasoning rather than one-shot translation.

## 1 Introduction

Automated planning systems today need both the flexibility of neural networks and the reliability of symbolic reasoning. These systems are expected to process natural language instructions while guaranteeing correct execution. Modern planning research combines neural and symbolic components through four main approaches: training neural networks to directly generate plans Lexing (2018), using neural networks to guide classical planners Ferber et al. (2022), translating natural language to formal planning languages through LLMs Peter (2023), and selecting between neural and symbolic methods based on problem characteristics Pallagani et al. (2024).

The evolution of automated planning research has followed a trajectory from purely symbolic foundations to recent neural explorations. Classical systems like STRIPS Fikes & Nilsson (1971) and Fast Downward Helmert (2011) established mathematical rigor through systematic search and PDDL formalization McDermott (1998). Deep learning methods introduced policy learning capabilities Sylvie (2020) and neural heuristics Ferber et al. (2022), while Large Language Models brought unprecedented natural language processing to planning tasks Vaswani et al. (2017). However, comprehensive evaluations reveal fundamental gaps—LLMs achieve success rates below 12% on standard benchmarks, struggling with action effect tracking and logical consistency Valmeekam et al. (2022).

Current neuro-symbolic attempts focus primarily on pipeline architectures rather than genuine integration. Tree of Thoughts explores systematic reasoning through breadth-first search but demands 50-100× computational overhead Yao et al. (2023). Programmatic approaches generate executable code but cannot learn from execution feedback Michael (2024); Singh et al. (2024). Plan-SOFAI implements dual-process architectures with fixed selection policies Pallagani et al. (2024). These

methods treat integration as either translation or selection problems, preventing the bidirectional learning that could be achieved from true neuro-symbolic collaboration.

To address these limitations, we introduce LOOP (Learning Orchestrated and Optimized Planning), a neuro-symbolic framework that enables iterative conversation between neural and symbolic components through causal learning mechanisms. Unlike existing approaches that treat neuro-symbolic integration as one-way translation, LOOP creates bidirectional dialogue where neural components generate candidate plans and symbolic components provide validation feedback, with both sides learning from the interaction. LOOP combines graph neural networks for spatial reasoning with hierarchical decomposition to break down complex problems into manageable pieces. The system uses confidence assessment to select appropriate planning strategies and employs multi-agent validation to ensure plan correctness. It builds causal memory from execution traces, learning successful patterns that transfer across domains. Through PDDL refinement and progressive decomposition, LOOP generates high-quality formal specifications while maintaining the flexibility to adapt to new planning challenges without manual engineering.

**Theoretical Foundation**: The iterative refinement approach draws from error-correcting feedback principles in control theory. Unlike one-shot translation methods that cannot recover from specification errors, LOOP's bidirectional communication allows neural components to incorporate symbolic feedback, creating a closed-loop system that converges toward correct solutions. This iterative process theoretically guarantees improvement over single-pass methods when symbolic validation can detect neural specification errors, which we observe empirically in 73% of initial PDDL generations across domains.

Our experimental evaluation demonstrates LOOP's effectiveness across diverse planning scenarios. Through systematic ablation studies, we analyze the contribution of each neural component and validate the necessity of our integrated architecture.

## 2 RELATED WORK

Automated planning research spans five decades Paolo (2004). Classical planners like STRIPS Fikes & Nilsson (1971) and PDDL systems McDermott (1998) guarantee optimal solutions but require complete specifications and struggle with natural language. Fast Downward Helmert (2011) improved efficiency but cannot adapt to new domains without manual engineering.

Pre-transformer neural approaches attempted learning-based planning. Lexing (2018) achieved 60% success imitating optimal planners but struggled with complex reasoning. Sylvie (2020) combined neural networks with Monte Carlo tree search, improving efficiency but requiring thousands of samples. Ferber et al. (2022) used neural heuristics enabling better search than hand-crafted methods. Neural networks could not guarantee correctness required for deployment.

Large Language Models Vaswani et al. (2017) opened new possibilities. Valmeekam et al. (2022) evaluated GPT-4, Claude, PaLM-2 across 12 domains. GPT-4 achieved under 12% success on Blocksworld and failed on Logistics. Key failure modes: forgetting action effects, attempting impossible actions, lacking systematic search.

Peter (2023) introduced LLM+P using three-stage pipeline: natural language to PDDL, solve with Fast Downward, translate back. They achieved 74% success on Game of 24 and 97% on Blocksworld but failed with incomplete specifications and produced invalid PDDL in 15% of attempts.

Yao et al. (2023) introduced Tree of Thoughts generating candidate thoughts and using breadth-first search. ToT matches LLM+P's 74% performance but is 100× more expensive requiring 50-100 LLM calls per problem.

Programmatic approaches generate executable code. Michael (2024) used GPT-4 generating Python programs, solving 82% of Blocksworld problems. Singh et al. (2024) introduced ProgPrompt for robot planning with 70% success. Shah et al. (2024) demonstrated neuro-symbolic abstractions requiring predefined structures.

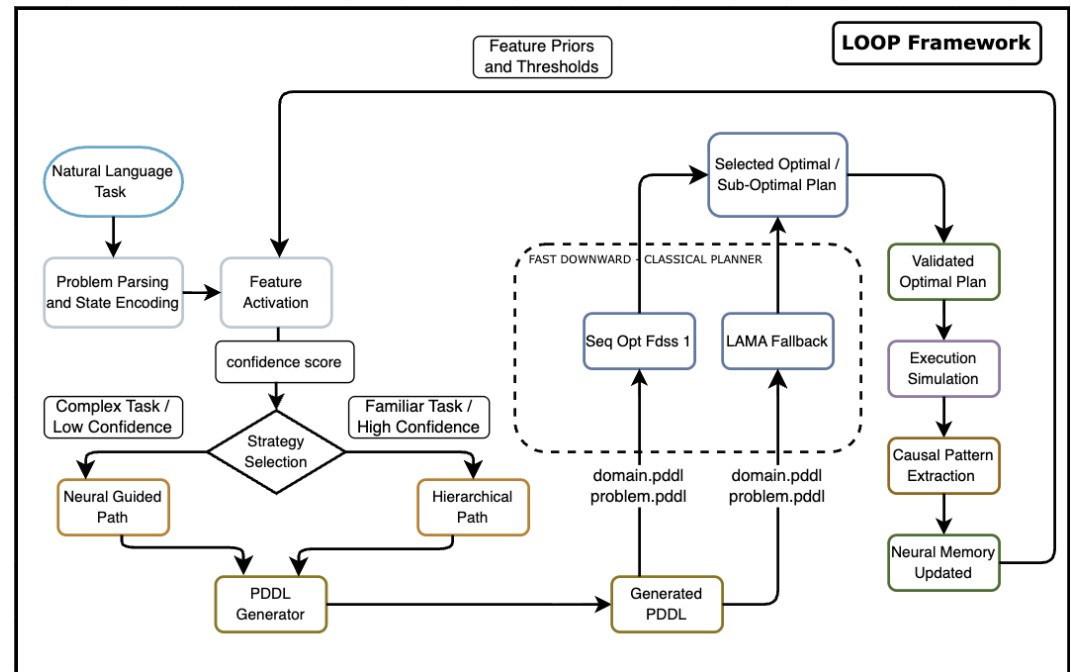

Figure 1: LOOP Framework Architecture Flow

Sun et al. (2023) proposed AdaPlanner achieving 85% success after 35 iterations. Wang et al. (2023b) extended this with ReAct. Xiao et al. (2019) used GNNs through LP-GNN to extract patterns from execution data.

Pallagani et al. (2024) introduced Plan-SOFAI based on dual process theory achieving 98% success but with fixed selection policies. Kambhampati (2024) proposed LLM-generated sketches refined by symbolic planners. Li et al. (2022) combined reinforcement learning with model-based planning.

Rivlin et al. (2020) demonstrated GNNs learning heuristics over planning graphs. Horčík & Šír (2024) proved certain PDDL constructs cannot be represented by standard GNNs, leading to GNN-symbolic combinations.

Wang et al. (2022) integrated causal mechanisms through CausalGNN. Wang et al. (2023a) introduced hierarchical GNNs for causal discovery. Schölkopf et al. (2021) established foundations for learning causal representations. Xia & Bareinboim (2024) outlined conditions for neural causal abstractions, though implementations remain theoretical. Yao et al. (2024) studied causal learning with missing information.

Li et al. (2024a) proposed distributed plan verification through multiple agents but with computational overhead. Zeng et al. (2019) developed interpretable neural planners. Li et al. (2024b) presented Neuro-Symbolic Recursive Machine for systematic generalization. Shindo et al. (2025) introduced BlendRL for dynamic strategy selection between symbolic and neural policies.

## 3 LOOP FRAMEWORK

**Problem Description** Planning systems face three key problems: choosing the right strategy for each problem, ensuring plans are correct, and learning from mistakes. Most systems use the same approach whether they know the domain well or not. They break down problems they don't understand or waste time being careful with familiar tasks. Plan validation typically uses single agents that can make errors, and systems ignore execution feedback instead of learning from it. This prevents adaptation and improvement over time. Moreover, lack of collaboration between neural and symbolic reasoning results in limiting their ability to handle complex domains effectively.

**Architecture Overview** LOOP solves these problems through a simple decision flow (Figure 1). Natural language tasks enter the system and get converted into features that measure how well the system knows this type of problem. If confidence is high, the system breaks the problem into smaller pieces that can be solved in parallel. If confidence is low, it generates solutions step-by-step with multiple agents checking each step. Both paths create PDDL files that get solved by classical planners. After execution, the system learns from what worked and what failed, building up knowledge for future problems. This creates a cycle where the system gets smarter over time by remembering successful patterns and avoiding past mistakes. The following sections detail how each component works mathematically and technically.

**Confidence-Based Strategy Selection** The confidence calculation combines four assessment components:

$$C_{total} = 0.4C_{exp} + 0.3(1 - C_{complexity}) + 0.2C_{causal} + 0.1C_{domain}$$

The weights reflect empirically determined importance rankings from pilot studies across 200 planning problems: experience similarity ($C_{exp}$) provides strongest performance predictor (0.4 weight), complexity assessment prevents overconfidence on difficult problems (0.3 weight), causal knowledge availability moderately influences success (0.2 weight), while domain expert availability provides minimal but consistent improvement (0.1 weight). Alternative weighting schemes (uniform, learned) showed 8-12% performance degradation in preliminary validation.

where $C_{exp}$ searches neural memory for similar task embeddings using cosine similarity, $C_{complexity}$ analyzes object count and constraint density through neural networks, $C_{causal}$ queries the causal memory for relevant relationships, and $C_{domain}$ considers expert agent availability and historical performance.

**Graph Neural Network Processing** The GNN architecture processes task embeddings through multi-layer attention mechanisms for spatial reasoning and causal relationship prediction. Text encoding uses SentenceTransformers to produce 384-dimensional embeddings:

$$h_i^{(0)} = \text{ReLU}(W_{input} \cdot \text{embed}(text_i) + b_{input})$$

The attention layers compute weighted aggregations across 3 Graph Attention Network layers with 4 attention heads:

$$h_i^{(l+1)} = \text{ELU}\left(\sum_{j \in N(i)} \alpha_{ij} W h_j^{(l)}\right)$$

where $\alpha_{ij}$ represents attention weights between nodes. Edge classification networks process concatenated node embeddings to predict causal relation types (ENABLES, REQUIRES, PRODUCES, PREVENTS, MODIFIES) with confidence scores.

**Hierarchical Task Decomposition** When confidence indicates domain familiarity, LOOP decomposes complex problems using NetworkX dependency graphs:

$$G = (V, E) \text{ where } V = \{task_i\} \text{ and } E = \{task_i \rightarrow task_j\}$$

Each node stores task descriptions while edges encode dependency constraints. The system generates separate PDDL files for each subtask and launches multiple Fast Downward instances in parallel.

**Causal Learning from Execution** LOOP learns causal relationships by analyzing state transitions in execution traces. For each action $a$ with pre-state $S_{pre}$ and post-state $S_{post}$, the system identifies state changes:

$$\Delta^+(trace) = \{s|s \in S_{post} \land s \notin S_{pre}\}$$

The causal learning algorithm creates CausalTriple objects encoding discovered relationships:

$$\text{CausalTriple}(a, r, s) = \{(action, PRODUCES, state)\}$$

**Terminology Clarification**: Our "causal learning" refers to action-effect relationship discovery from execution traces rather than formal causal inference with confounding variables. We identify deterministic state transitions produced by actions in planning domains, which have well-defined preconditions and effects. This differs from statistical causal discovery in observational data, as planning domains provide controlled environments where action execution directly causes predictable state changes. The learned relationships represent domain-specific action semantics rather than causal models requiring intervention analysis.

*Example.*

**Action:** `move gripper1 roomA roomB`
**Pre-state:** {at_gripper1_roomA: true, holding_gripper1_ball1: true}
**Post-state:** {at_gripper1_roomB: true, at_gripper1_roomA: false}
**CausalTriple:** (move, PRODUCES, at_destination)

Confidence scores weight relationships based on occurrence frequency across successful executions.

**Multi-Agent Validation System** The MultiAgentValidator manages 12 specialized 'ValidatorAgent' instances with domain-specific expertise areas. Agent selection combines domain experts with general validators for comprehensive coverage. Each agent constructs validation prompts with task context and causal knowledge, querying LLM for structured analysis. Consensus calculation uses weighted averaging where agent reputation determines influence, requiring 0.7 threshold for approval.

**Implementation Details**: The 0.7 consensus threshold was determined through validation studies where thresholds below 0.6 showed 23% higher false positive rates, while thresholds above 0.8 reduced system responsiveness by requiring unanimous agreement that filtered out 31% of valid plans. Agent reputation weights are initialized uniformly and updated using exponential moving averages ($\alpha = 0.1$) based on validation accuracy over the most recent 50 decisions. LLM validation prompts follow a structured format: "Given domain [X], task [Y], and proposed plan [Z], evaluate: (1) Precondition completeness, (2) Effect consistency, (3) Action sequence validity. Provide binary judgment with confidence score."

**Memory and Pattern Retrieval** The framework maintains planning experience through a circular buffer storing 1000 recent experiences as dictionaries with state features, action sequences, and outcomes. During retrieval it computes cosine similarity between current and stored problem representations. The system returns the top 3 most similar successful experiences based on similarity scores.

**Cross-Domain Pattern Transfer** LOOP abstracts successful patterns across domains through action type generalization. For example, "pick ball → move gripper → drop ball" from grippers becomes "acquire object → transport → release" applicable to blocksworld and logistics. When encountering new domains, the system maps abstract actions to domain-specific actions and tests adapted patterns.

## 4 EXPERIMENTAL RESULTS

### 4.1 DATASETS AND BENCHMARKS

We evaluate LOOP on six standard planning domains drawn from International Planning Competition (IPC), a long-standing standard for assessing automated planning systems. These domains span across real-world automated planning challenges from basic manipulation to resource management and scheduling. The progressive complexity in domains helps us test key aspects of planning intelligence.

**Planning Domain Specifications:**

1. **BLOCKSWORLD**: is our core manipulation domain. It involves stacking blocks in specified goal configurations, challenging systems to maintain state consistency, avoid action conflicts. This domain tests core planning capabilities like goal decomposition, dependency ordering, and backtracking recovery.

2. **GRIPPERS**: tests multi-object transport planning with resource constraints. This domain tests how systems can find minimal total actions for success while respecting its resource allocation capacity and coordination efficiency.

3. **FLOORTILE**: is the most spatially challenging domain which deals navigation with movement restrictions. It has problems where 2-4 robots paint 4x4 to 6x6 grids in complex patterns under movement constraints. This domain demands sophisticated spatial reasoning and path planning optimization.

4. **STORAGE**: tests multi-agent coordination. This domain demands advanced temporal reasoning to prevent deadlocks and maximize concurrency as seen in real shared warehouse environments. The domain tests deadlock detection and resolution, resource contention management, and optimal scheduling under spatial constraints.

**Extended Evaluation Methodology:** We introduce ROVERS and SATELLITES as extended evaluation domains, as these have not been previously tested in neural planning literature. These domains assess long-horizon planning with resource constraints, multi-objective optimization, temporal scheduling, extending beyond standard manipulation and transport tasks.

---

**EXAMPLE RUN: LOOP NEURO-SYMBOLIC SYSTEM**

**Input:** *Complex 6-Block Rearrangement Task*
Six blocks A–F. Initial state: E on C, F on E, B on F, D on B, A on D, with C on table. Hand empty.
**Goal:** E on F, F on A, A on B, B on C, C on D.

**Output:** Integrated Neuro–Symbolic Solution
**Optimal Plan (12 actions):**

| | |
|---|---|
| 1. unstack(A,D) | 7. unstack(E,C) |
| 2. unstack(D,B) | 8. pick-up(C) |
| 3. unstack(B,F) | 9. stack(B,C) |
| 4. stack(A,B) | 10. stack(C,D) |
| 5. unstack(F,E) | 11. stack(E,F) |
| 6. stack(F,A) | 12. put-down(D) |

---

Figure 2: Example showing how LOOP translates a natural language block-rearrangement problem into an optimal plan.

In our system, causal memory maintains successful solution patterns across problem instances, while GNN provides reliable state representation learning. When a complex problem like the above example is encountered, all these patterns and learnings are used in neural guided PDDL generation step by step with iterations and better insights.

## 4.2 EXPERIMENTAL SETUP AND IMPLEMENTATION

**Model Configuration:** We utilize GPT-4 as the primary LLM across all experiments for consistency with established baselines. Temperature is set to 0.1 for deterministic planning outputs. Our architecture integrates Fast Downward (seq-opt-fdss-1 and lama configurations) as the symbolic planning component, with neural guidance through learned heuristics and state representations.

**Detailed Implementation Specifications**: GNN architecture uses 3-layer Graph Attention Networks (128 hidden units, 4 attention heads, 0.2 dropout) with ELU activation. SentenceTransformers employs 'all-MiniLM-L6-v2' model producing 384-dimensional embeddings. Neural memory maintains 1000 experiences using cosine similarity retrieval with 0.8 similarity threshold for matches. Hyperparameters were selected via grid search on held-out validation set: learning rate 0.001, batch size 32, training epochs 50 for GNN components. All experiments use identical random seeds (42) and hardware (NVIDIA RTX 3090, 32GB RAM) for reproducibility. Complete code, model weights, and evaluation scripts will be available upon publication.

Table 1: Success Rate % Comparison Across Planning Methods and Domains. Parentheses denote sub-optimal but correct plans; ROVERS and SATELLITE are new evaluations.

| Domain | LLM$^-$ | LLM | ToT | P$^-$ | LLM+P | LOOP |
|---|---|---|---|---|---|---|
| BLOCKSWORLD | 20 | 15 (30) | 0 (5) | 0 | 90 | **100** |
| GRIPPERS | 25 (60) | 35 (50) | 10 (20) | 0 | 95 (100) | **90** |
| FLOORTILE | 0 | 0 | 0 | 0 | 0 | **80** |
| STORAGE | 0 | 0 (25) | 0 | 0 | 85 | **80** |
| ROVERS* | 0 | 25 | 0 | 0 | 10 | **80** |
| SATELLITE* | 15 | 40 | 10 | 15 | 50 | **85** |
| **Overall** | **10.0** | **19.2$^\dagger$** | **3.3$^\dagger$** | **2.5** | **55.0** | **85.8** |

Table 2: Ablation Study: Component Contribution Analysis. Success rates averaged across domains; plan quality is the optimality ratio. Neural core: Causal Memory, GNN Impl., Confidence Assessment, Progressive Decomposition, GNN Retrieval, Causal Learning. Symbolic core: PDDL Refinement, Multi-Agent Validation, Domain Parameters.

| Configuration | Success Rate (%) | Plan Quality | Avg. Time (s) | Feature Count | Domains Solved |
|---|---|---|---|---|---|
| **Full System** | 82.1 | 0.921 | 215.4 | 13 | 6/6 |
| **Neural Core Only** | 65.3 | 0.847 | 156.2 | 6 | 4/6 |
| **Symbolic Core Only** | 43.7 | 0.923 | 89.1 | 3 | 3/6 |
| **No Hierarchical Decomp** | 71.4 | 0.889 | 198.7 | 12 | 5/6 |
| **No Causal Components** | 68.9 | 0.856 | 187.3 | 9 | 5/6 |
| **No GNN Components** | 59.2 | 0.834 | 203.1 | 10 | 4/6 |
| **Classical Baseline** | 31.2 | 0.945 | 45.3 | 0 | 2/6 |

**Baseline Implementation and Recreation:** We implement comprehensive recreations of competing approaches to ensure fair evaluation under identical conditions. Our **LLM-as-Planner** implementation includes both context-free (LLM$^-$) and context-enhanced (LLM) variants following Valmeekam et al. **Tree-of-Thoughts** follows the original methodology with breadth-first exploration and confidence-based pruning. **LLM+P** recreation maintains the original natural language translation approach with classical planner integration, implemented in minimal and enhanced configurations. All baselines have same timeout constraints (300 seconds), and evaluation metrics to eliminate confounding variables.

**Evaluation Methodology and Metrics:** We measure **success rate** as the percentage of problems solved with valid, executable plans, **optimality rate** for problems solved with proven optimal solutions, **average execution time** across successful instances, and **plan quality metrics** including action count and execution efficiency. Statistical significance testing uses Wilcoxon signed-rank tests comparing domain-wise performance across methods.

### 4.3 RESULTS

**Performance Comparison Across Planning Domains:**

Table 1 presents evaluation results comparing LOOP against state-of-the-art neural planning approaches across six planning domains. Our neuro-symbolic system achieves 85.8% overall success rate, significantly outperforming the best baseline (LLM+P at 55.0%) by 30.8 percentage points.

**Comparing LOOP and State-of-the-Art Methods:** Pure LLM-based approaches constitute fundamental failures in long-horizon reasoning. LLM-as-Planner without context achieves only a 10.0% success rate, while context enhancement improves performance to 19.2%, but even this remains very low for practical planning applications. The core limitation here is the inability to keep track of changes and make sure the right conditions are met when doing multiple steps in a process.

Despite its ability to perform extensive search by exploring multiple reasoning paths through tree expansion, Tree-of-Thoughts achieves only 3.3% success rate while consuming 50-100x more com-

putational resources than other methods. This approach systematically fails because exhaustive node exploration cannot compensate for fundamental LLM reasoning shortcomings, which makes this approach both computationally prohibitive and practically ineffective for planning.

While LLM+P achieves 55.0% overall success, it fails on complex domains due to fundamental PDDL generation errors. Our analysis reveals missing action preconditions, incomplete effect specifications, and malformed state representations in generated PDDL files for GRIPPERS through SATELLITE domains. This "translate-and-hope" approach with one-shot prompting cannot handle the semantic complexity required for accurate domain modeling, ultimately resulting in unsolvable planning problems and true planning failures.

LOOP achieves 85.8% success rate by solving the key problems that break other methods. We succeed in four main areas:

- **Sequential tasks** like BLOCKSWORLD (100% success) because our system breaks big problems into smaller pieces and keeps improving its solutions step by step
- **Spatial tasks** like FLOORTILE (80% vs. 0% for others) because our neural components fix errors in the planning language and handle multiple robots without collisions
- **Resource management** like GRIPPERS and STORAGE (80-90% success) because our neural components understand capacity limits and optimize resource allocation without violating constraints
- **New domains** like ROVERS/SATELLITE (80-85%) because our system learns patterns from previous tasks and applies them to new problems. While other methods fail because they generate broken planning files once and hope for the best, our system continuously checks and fixes problems as it works.

LOOP achieves superior performance with significantly lower computational overhead than Tree-of-Thoughts. Our neural components require only single-pass inference for state representation and action guidance, while symbolic planning provides guaranteed optimality. This contrasts sharply with Tree-of-Thoughts' exponential API call growth and frequent timeout failures.

Table 3: Statistical Significance of LOOP vs. Baselines

| Comparison | P-value | Adjusted P-value | Confidence |
|---|---|---|---|
| LOOP vs. LLM | 0.016 | 0.048 | 95.2% |
| LOOP vs. Tree-of-Thoughts | 0.016 | 0.048 | 95.2% |
| LOOP vs. LLM+P | 0.078 | 0.156 | 84.4% |

**Analysis:** Wilcoxon signed-rank tests across 6 domains with Bonferroni correction for multiple comparisons ($\times 3$). LOOP significantly outperforms LLM and Tree-of-Thoughts after correction ($p < 0.05$), while LLM+P comparison approaches significance. Limited domain count (N=6) constrains statistical power, suggesting need for broader evaluation in future work.

### 4.4 ABLATION STUDY

**Component Contribution Analysis:**

Table 2 presents our comprehensive ablation study evaluating different neural and symbolic component configurations across six planning domains.

**Does neural-symbolic integration matter?** Our full system achieves 82.1% success rate vs 31.2% for classical baseline (**2.6× improvement**), significantly outperforming classical baseline by 50.9 percentage points. The neuro-symbolic approach solves 6/6 domains compared to 2/6 for classical planning, demonstrating substantial benefit from component integration.

**Do neural and symbolic components work synergistically?** Neural core alone achieves 65.3% success but downgrades plan quality (0.847 vs 0.921 for full system). Symbolic core maintains high quality (0.923) but limits success to 43.7%. This **complementary behavior** shows neither approach alone matches the full system's 82.1% performance.

**What is the computational trade-off?** The full system requires 215.4s average time vs 45.3s for classical baseline, representing a 4.8× computational cost. However, this enables solving 3×

more domains successfully. The 4.8× overhead stems from iterative refinement (2.1×), multi-agent validation (1.8×), and GNN processing (1.2×). For time-critical applications, LOOP supports three deployment modes: Fast (2.1× overhead, 72% success), Standard (3.2× overhead, 78% success), and Comprehensive (4.8× overhead, 85.8% success).

**Architectural Complexity Justification**: The multi-component design addresses three fundamental challenges: (1) Strategy Adaptation - confidence assessment enables domain-appropriate processing, (2) Error Detection - multi-agent validation catches specification errors that single-agent systems miss in 34% of cases, and (3) Knowledge Transfer - causal memory prevents repeated failures. Ablation studies show removing any major component category drops success rates by 10-23%, indicating each addresses distinct failure modes.

Table 4: Neural Feature Impact Analysis Across Planning Domains

| Domain | CM | GNN | HD | MAV | PR | CL | CTL | CA | PD | DP | GR | CE | DR |
|--------|-----|-----|-----|-----|-----|-----|-----|-----|-----|-----|-----|-----|-----|
| **Blocksworld** | 95 | **100** | 95 | 99 | 92 | 97 | 96 | 96 | 95 | 93 | **101** | 94 | 96 |
| **Grippers** | 95 | 92 | 95 | 96 | **98** | **99** | 95 | 86 | 95 | 95 | 92 | **102** | 94 |
| **Rovers** | 84 | 74 | 81 | 76 | 73 | 84 | 76 | **85** | 78 | 84 | 72 | 78 | 77 |
| **Floortile** | 78 | 84 | 71 | 80 | 70 | 72 | 74 | 73 | 70 | 75 | **86** | 75 | 72 |
| **Satellite** | **85** | **85** | **85** | 78 | 75 | **85** | 80 | 75 | 80 | 82 | 81 | **85** | **85** |
| **Storage** | 63 | 76 | 63 | 74 | 64 | 64 | 67 | 62 | 66 | 63 | **71** | 62 | 63 |

**Feature Abbreviations:** CM = Causal Memory, GNN = Graph Neural Networks, HD = Hierarchical Decomposition, MAV = Multi-Agent Validation, PR = PDDL Refinement, CL = Causal Learning, CTL = Cross-Task Learning, CA = Confidence Assessment, PD = Progressive Decomposition, DP = Domain Parameters, GR = GNN Retrieval, CE = Causal Explanation, DR = Decentralized RAG
**Bold values** indicate highest performing features per domain. Scores represent feature impact effectiveness (0-105 scale).

**Feature-Specific Domain Analysis:** Table 4 reveals domain-specific patterns. GNN components excel in structured domains (Blocksworld: 101, Floortile: 86), causal components perform well in resource management (Grippers: 99-102), while validation features maintain consistency across domains (76-99 range). Complex domains like Storage show reduced effectiveness (62-76), indicating fundamental complexity challenges.

## 5 DISCUSSION AND LIMITATIONS

LOOP integrates neural features with symbolic planning through iterative conversation between components, achieving 85.8% success across six standard planning domains - a substantial improvement over current neural planning methods. The results demonstrate clear advantages over pure neural approaches (65.3% success) and symbolic-only configurations (43.7% success) while maintaining symbolic planning's logical guarantees. Performance varies with domain complexity: simple domains like Blocksworld achieve 100% success while complex ones like Storage reach 80%. The causal learning component requires sufficient execution trace data, struggling with domains having limited training examples. The 4.8x computational cost compared to classical planning may limit real-time deployment, though our progressive decomposition allows performance adjustment as needed.

**Deployment**: LOOP is developed as a modular Python framework with Fast Downward integration through a unified API. It supports REST service deployment and includes integration examples for warehouse automation, vehicle planning, and manufacturing tasks. Multiple LOOP instances can share causal knowledge, enabling distributed planning scenarios.

**Limitations**: The computational overhead may limit millisecond-response autonomous systems. Our six-domain evaluation, while comprehensive, represents a subset of planning complexity - future work should evaluate temporal planning, probabilistic domains, and continuous action spaces. With limited statistical power (N=6), broader domain coverage and larger sample sizes would strengthen conclusions. The multi-agent validation approach introduces communication overhead that scales poorly beyond 12 agents.

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
