# OpenReview forum: "LOOP: A Plug-and-Play Neuro-Symbolic Framework for Enhancing Planning in Autonomous Systems"
_ICLR.cc/2026/Conference — ICLR 2026 Conference Withdrawn Submission_

### Official Review · Reviewer_1UBs · 2025-10-31

**Soundness:** 3
**Presentation:** 2
**Contribution:** 2
**Rating:** 2
**Confidence:** 5

**Summary:**

The paper introduces LOOP, a plug-and-play neuro-symbolic framework designed to enhance automated planning in autonomous systems. LOOP enables iterative interaction between neural and symbolic modules, integrating causal learning, multi-agent validation, and hierarchical decomposition to progressively refine PDDL-based plans. The framework models planning as a closed-loop process, allowing neural components to generate candidate plans while symbolic planners provide feedback for correction and convergence. Evaluated on six IPC domains and two extended benchmarks, LOOP achieves an average success rate of 85.8%.

**Strengths:**

- LOOP combines neural reasoning, symbolic validation, and causal learning into a cohesive iterative pipeline that addresses key limitations of one-shot LLM planning.
- The neuro-symbolic interaction and causal trace learning offer a degree of transparency uncommon in LLM-integrated planners.

**Weaknesses:**

- It would be helpful to compare against PSALM [1], which reports higher success rates on the same IPC domains using environment feedback for action semantics induction.
- The paper omits several relevant works from Kambhampati’s group, including Guan et al. [2] and others on feedback-guided planning and world-model construction, which should be cited and discussed for completeness.
- Many tables are oversized and difficult to read in the current layout, consider splitting them across pages or reformatting for clarity.
- The evaluation primarily focuses on GPT-4; it remains unclear whether similar performance can be achieved with smaller or open-weight models, which affects reproducibility and accessibility.
- How does LOOP handle partial or noisy feedback, which is essential for demonstrating the benefits of causal memory and multi-agent validation?

References:
- [1] Zhu et al., Language Models can Infer Action Semantics for Classical Planners from Environment Feedback (NAACL 2025).
- [2] Guan et al., Leveraging Pre-trained Large Language Models to Construct and Utilize World Models for Model-based Task Planning (Neurips 2023).

**Questions:**

See weaknesses.

---

### Official Review · Reviewer_pVHv · 2025-11-01

**Soundness:** 2
**Presentation:** 1
**Contribution:** 2
**Rating:** 0
**Confidence:** 4

**Summary:**

This paper introduces LOOP, a neuro-symbolic planning framework that models planning as an iterative interaction between neural and symbolic modules. LOOP demonstrates good performance across six benchmark domains, outperforming several baseline methods.

**Strengths:**

The proposed method achieves good performance on 6 domains.

**Weaknesses:**

1. The paper is poorly written and contains several unreasonable or problematic aspects:

(1) The authors claim that their method can refine PDDL models, but the paper does not clearly explain how this refinement is actually performed.

(2) The Related Work section lacks a sufficient discussion distinguishing the proposed approach from existing methods.

(3) The Theoretical Foundation section fails to provide the necessary theoretical background or analysis to justify the effectiveness of the proposed iterative optimization method.

(4) The paper suffers from noticeable formatting issues:
  (a) Tables 2 and 4 exceed the normal paragraph spacing;
  (b) Quotation marks are incorrectly used. The left double quotation mark should be `` and the left single quotation mark should be `;
  (c) Example (Line 228-232) is inserted directly into the main paragraph without following the proper paragraph formatting style.

2. The evaluation only considers GPT-4 as the underlying LLM in the proposed system. Without testing additional language models, it is difficult to assess the general applicability and robustness of the proposed framework.

**Questions:**

See weaknesses.

---

### Official Review · Reviewer_cSaU · 2025-11-02

**Soundness:** 1
**Presentation:** 1
**Contribution:** 1
**Rating:** 0
**Confidence:** 5

**Summary:**

This paper tries to explain an implementation of a LLM-based (sentence-transformer embedding) system that
supposedly uses classical planners as validators that provide a feedback to LLM,
a common approach in the recent natural-language-based planning literature.

While the abstract claims that it achieves a good performance,
it is unfortunately immediately obvious that
this paper is either not proofread enough or completely written by LLMs,
making it not ready for submission or publication.

**Strengths:**

None.

If there is anything that caught my eye,
it is the idea of Cross-Domain Pattern Transfer;
It effectively stores a macro-action as a policy,
but in a way that generalizes the macro action with natural language semantics.
This sounds new and interesting,
but the paper in the current state cannot is not publishable.

**Weaknesses:**

The writing issue starts from the very first page of the paper.
Having so many issues in the very face of the paper (= introduction) means that this paper has not be thoroughly proofread.
Issues that are immediately spotted are

-   Many \cite / \citet / \citep command misuses in the introduction.
-   Many style inconsistency in the references, e.g.,
    -   ICAPS conference proceedings are sometimes with the number (Thirty-Fourth &#x2026;) and sometimes not
    -   Citations without venues (e.g. just "2024a")
-   Overfull hbox in many tables

The style in which the main text explains the method (Section 3) is so unorthodox, almost ChatGPT-like, and reads completely gibberish to me.
The explanation of the main method is so abrupt with so little justification for the design choice,
leaving little room for the readers to gain meaningful insights.

First of all, what is the task, what is the input, what is the output in Figure 1? Formalize it.
Then, going over the text, more concretely,

> LOOP solves these problems through a simple decision flow (Figure 1).

Figure 1 is NOT simple at all!

> Natural language tasks enter the system and get converted into features that measure how well the

What features? How it is converted?

> system knows this type of problem. If confidence is high, the system breaks the problem into
> smaller pieces that can be solved in parallel. If confidence is low, it generates solutions step-by-

How does it decompose the problem? Note that this is possible only for a select subset of planning domains (serial decomposability),
and domains with serial decomposability are considered easy. In other words, this system works only on easy tasks.

> step with multiple agents checking each step. Both paths create PDDL files that get solved by
> classical planners. After execution, the system learns from what worked and what failed, building
> up knowledge for future problems. This creates a cycle where the system gets smarter over time
> by remembering successful patterns and avoiding past mistakes. The following sections detail how
> each component works mathematically and technically.

What is the output?

> Confidence-Based Strategy Selection

> The weights reflect empirically determined importance rankings from &#x2026;

How did you determine the coefficients?

-   experience similarity (Cexp)
-   complexity assessment
-   causal knowledge availability
-   domain expert availability

None of them are defined yet.

> where Cexp searches neural memory for similar task embeddings using cosine similarity,
> Ccomplexity analyzes object count and constraint density through neural networks, Ccausal queries
> the causal memory for relevant relationships, and Cdomain considers expert agent availability and
> historical performance.

First of all, this paragraph begins with an incomplete sentence.

-   neural memory
-   object count and constraint density
-   causal memory for relevant relationships
-   expert agent availability

None of these are defined yet.

> causal relation types (ENABLES, REQUIRES, PRODUCES, PREVENTS, MODIFIES)

Not explained at all.

> Hierarchical Task Decomposition

The paper does not explain how it decomposes the task at all.

I do not believe it is worth roasting the entire reminder of the paper this way,
which resembles the bullet-point style output from ChatGPT.

Being the LLM-produced nature of this paper,
I cannot reasonably trust the authors that they did not produce the tables using LLM.
This renders the entire empirical section and the numbers in the table unreliable, making the review meaningless.

**Questions:**

None. Already explained above.

---

### Note · Authors · 2025-11-12

I have read and agree with the venue's withdrawal policy on behalf of myself and my co-authors.